# Examination of Wnt signaling as a therapeutic target for pancreatic ductal adenocarcinoma (PDAC) using a pancreatic tumor organoid library (PTOL)

Hayley J. Hawkins[1]*, Betelehem W. Yacob[1], Monica E. Brown[2], Brandon R. Goldstein[2], John J. Arcaroli[1], Stacey M. Bagby[1], Sarah J. Hartman[1], Morgan Macbeth[1], Andrew Goodspeed[3,4], Thomas Danhorn[3,4], Robert W. Lentz[1], Christopher H. Lieu[1], Alexis D. Leal[1], Wells A. Messersmith[1], Peter J. Dempsey[2], Todd M. Pitts[1]

1 University of Colorado Anschutz Medical Campus, Aurora, CO, United States of America, 2 Section of Developmental Biology, Dept. of Pediatrics, University of Colorado Anschutz Medical Campus, Aurora, CO, United States of America, 3 University of Colorado Comprehensive Cancer Center, University of Colorado Anschutz Medical Campus, Aurora, CO, United States of America, 4 Department of Biomedical Informatics, University of Colorado Anschutz Medical Campus, Aurora, CO, United States of America

* hhawkins9417@gmail.com

**Data Availability Statement:** All data files are available from figshare database, with DOI 10.6084/m9.figshare.25035821.

## Abstract

Pancreatic ductal adenocarcinoma (PDAC) presents at advanced stages and is refractory to most treatment modalities. Wnt signaling activation plays a critical role in proliferation and chemotherapeutic resistance. Minimal media conditions, growth factor dependency, and Wnt dependency were determined via Wnt inhibition for seven patient derived organoids (PDOs) derived from pancreatic tumor organoid libraries (PTOL). Organoids demonstrating response *in vitro* were assessed *in vivo* using patient-derived xenografts. Wnt (in)dependent gene signatures were identified for each organoid. Panc269 demonstrated a trend of reduced organoid growth when treated with ETC-159 in combination with paclitaxel or gemcitabine as compared with chemotherapy or ETC-159 alone. Panc320 demonstrated a more pronounced anti-proliferative effect in the combination of ETC-159 and paclitaxel but not with gemcitabine. Panc269 and Panc320 were implanted into nude mice and treated with ETC-159, paclitaxel, and gemcitabine as single agents and in combination. The combination of ETC-159 and paclitaxel demonstrated an anti-tumor effect greater than ETC-159 alone. Extent of combinatory treatment effect were observed to a lesser extent in the Panc320 xenograft. Wnt (in)dependent gene signatures of Panc269 and 320 were consistent with the phenotypes displayed. Gene expression of several key Wnt genes assessed via RT-PCR demonstrated notable fold change following treatment *in vivo*. Each pancreatic organoid demonstrated varied niche factor dependencies, providing an avenue for targeted therapy, supported through growth analysis following combinatory treatment of Wnt inhibitor and standard chemotherapy *in vitro*. The clinical utilization of this combinatory treatment modality in pancreatic cancer PDOs has thus far been supported in our patient-derived xenograft models treated with Wnt inhibitor plus paclitaxel or gemcitabine. Gene expression analysis suggests there are key Wnt genes that contribute to the Wnt (in)dependent

**Funding:** The Wings of Hope for Pancreatic Cancer Research Pilot Grant. The award number was Wings.2019.004. Additionally, the study was partly supported by the National Institutes of Health P30CA046934 Bioinformatics and Biostatistics Shared Resource and Organoid and Tissue Modeling Shared Resource (support grant awarded to the University of Colorado Cancer Center). The funders had no role in study design, data collection and analysis, decision to publish, or preparation of the manuscript.

**Competing interests:** The authors have declared that no competing interests exist.

phenotypes of pancreatic tumors, providing plausible mechanistic explanation for Wnt (in) dependency and susceptibility or resistance to treatment on the genotypic level.

## Introduction

Pancreatic ductal adenocarcinoma (PDAC) is currently the fourth most common cause of cancer death with a 5-year survival rate of less than 8% due to the tendency of presentation at an advanced stage while being refractory to most treatment modalities [1]. Many mutations and somatic copy number alterations (SCNAs) in key oncogenes and tumor suppressor genes, such as KRAS, TP53, SMAD4, and CDKN2a, lead to the development of PDACs [2, 3]. In addition to these genetic alterations, the Cancer Genome Atlas Research Network has revealed an even more complex molecular landscape of PDACs and chromosomal rearrangement patterns [4]. This highlights the difficulties that arise when assessing the relationship between the genotype and biological responses of tumors The low tumor cellularity and dense stroma commonly found in pancreatic cancers further convolute these relationships and serve as major barriers to accurate and effective preclinical studies [5, 6].

Previous work by Sato and Clevers successfully developed *in vitro* 3D organoid cultures from gastrointestinal tissues. This technology has been expanded to propagate a variety of other endodermally-derived epithelial tissues, including the pancreas, through the utilization of appropriate stem cell niche factors [1, 7]. Importantly, tumor organoids generated from PDAC patients maintain histological traits, molecular characteristics, and genetic-mutation profiles of the parental tumors [1, 8]. Several groups have established PDAC tumor organoid libraries (PTOLs) [1, 8–11] and are using these new *in vitro* models as a more accurate platform to assess the relationships between cancer genotype and phenotype [7].

Wnt signaling is embryonically essential during pancreatic development, though its role is limited in a normal adult pancreas [12]. However, in response to pancreatic duct ligation injury, these cells acquire a progenitor-like state that reactivates Wnt signaling [13–16]. Consistent with Wnt signaling reactivation in response to pancreatic injury, both Wnt and R-Spondin are essential niche factors required for growth of normal pancreatic ductal organoids. While this dependency is seen in normal pancreatic ductal organoids, previous work done by the Sato group found that PDAC organoids display diverse requirements for Wnt signaling [1]. However, without specific Wnt pathway gene driver mutations (APC, AXIN2, CTNNB1, RNF43, etc.), it is not possible to predict Wnt/R-Spondin dependency through genetic mutational profiling or gene expression studies [1].

Currently, PDAC is typically managed surgically with chemotherapeutic options such as gemcitabine, nab-paclitaxel, and FOLFIRINOX if surgical resection is ineffective or not feasible [17–20]. Even with systemic chemotherapy, these tumors are often too advanced or resistant to treatment for effective tumor burden reduction further emphasizing the need for additional treatment modalities [21]. Significantly, the discovery of different modes of Wnt (in)dependency in PDAC gives rise to the question of a novel Wnt-based therapy for Wnt-dependent tumors, as is being investigated in colorectal cancer patients in various clinical trials [22].

The Wnt pathway not only contributes to the growth and maintenance of cancer stem cells through the suppression of p21Cip1, it also increases the tolerance of DNA damage by upregulating survivin [23, 24]. Survivin inhibits apoptosis and is a contributing factor to gemcitabine resistance [21, 25]. In colon, prostate, and breast cancers, inhibition of β-catenin enhances the

cytotoxicity of standard chemotherapy. Given the increased propagation of cancer cells and chemotherapeutic resistance observed by WNT pathway activation, targeting of the Wnt pathway is a viable option to overcome the barriers of chemotherapeutic resistance. Pancreatic cancer patients treated with the human monoclonal antibody vantictumab that binds Frizzled receptors demonstrated promising results in combination with standard therapy in a Phase 1b study with an overall response rate of 42% [15, 26, 27]. Alternate targets of the canonical and non-canonical Wnt pathway have included the Porcupine (PORCN) inhibitor, ETC-159. In several clinical trials ETC-159 has shown promising efficacy in pancreatic cancer cell lines *in vivo* with significant tumor growth inhibition and no tumor regrowth following drug removal [1, 15].

Although Wnt activating mutations are rare in PDACs, recent observations by Sato's group and our own preliminary data using PTOLs to identify Wnt dependent PDAC subtypes suggest that Wnt dependency can be informative in therapeutic treatment strategies to overcome chemotherapy resistance. Thus, the goal of this study was to determine whether Wnt dependency will predict Wnt inhibitor response and lead to reversing chemotherapy resistance.

## Methods

### Human specimens

Patient derived tumor samples were collected from consenting PDAC patients undergoing surgical resection or tumor biopsy at the University of Colorado Cancer Center with approval by the Colorado Multiple Institutional Review Board (08–0439). Tissue samples were confirmed to be tumor or normal based on pathologist assessment. Authors were blinded and had no access to identifying patient information. Patient tumor mutation data was obtained from archived tumor samples and prepared as described below for whole exome sequencing.

### PDAC patient-derived xenografts

All animal work was performed with approval by the University of Colorado Anschutz Medical Campus IACUC. All personnel were trained in animal care, dosing, tumor studies, and handling prior to animal studies and submitted personnel qualification forms to the IACUC. Fresh PDAC tissue samples were used to generate patient-derived xenograft (PDX) models as described previously [28].

### Pancreatic Tumor Organoid Library (PTOL) generation and characterization

**PTOL generation.** All human PDAC organoid lines were generated from established patient-derived xenograft (PDX) PDAC tumor models or from fresh primary PDAC tumor tissue as previously described with some modifications [1, 29]. To date, 8 PDAC organoid lines have been established (S1 Table). Six PDAC organoid lines were generated from PDX PDAC tumors (Panc129, Panc193, Panc269, Panc271, Panc272, and Panc320) whereas two lines were generated directly from primary pancreatic tumor tissue (Panc308 and Panc368). Briefly, PDAC tumor tissue was washed vigorously and minced with surgical scissors. The fragments were then digested with digestion media [Advanced DMEM/F12 (Gibco) containing 1% fetal bovine serum (FBS), 10mM HEPES (Gibco), 2mM GlutaMax (Gibco) and 1% Penicillin/Streptomycin (Gibco) with 1.5mg/ml collagenase (C2139 Sigma), 125ug/ml dispase, 0.1mg/ml DNAse1 (EMD Millipore) and 10μM Y27632] at 37˚C for 30 min. The digested cells were washed with wash media (DMEM containing 10% FBS, 10mM HEPES, 2mM Glutamax, 1% penicillin/streptomycin and 10μM Y27632) to inactivate the digestive enzymes, and then

filtered through a 100μm strainer to removed large undigested fragments prior to culture. Isolated pancreatic cells were embedded in Basement Membrane Extract (BME, Biotechne 9mg/ml) and medium was refreshed every 2–3 days.

**PTOL characterization.** Basal culture media contained advanced DMEM /F12 supplemented with 1% penicillin/streptomycin, 10mM HEPES, 2 mM GlutaMAX, 1 X B27 (Invitrogen), 10 nM gastrin, 1mM N-acetylcysteine and 10mM nicotinamide. Human Pancreatic Stem Cell (HPSC) media contained basal culture media supplemented with 50 ng/ml mouse recombinant EGF, 50% Wnt3A, Noggin, R-Spondin-3 (WNR) conditioned media, 500 nM A83-01 (Tocris), 10 μM SB202190 (Sigma) and 100 mg/ml Primocin (Vitrogen). Organoid media was refreshed every two days and organoids were passaged as needed based on organoid growth and confluency. For the first 2 days after passaging, organoid media was supplemented with 10 uM Y27632. Frozen stocks of all PDAC organoid lines were prepared and all experiments were performed within a 10–20 passage window. Organoid cultures were routinely tested for Mycoplasma.

## Assessment of the Wnt (in)dependency of PDAC tumor organoids

**HPSC media used for niche factor dependency studies.** HPSC media is comprised of the WENRAS niche factors: Wnt3A (W), EGF (E), Noggin (N), R-Spondin (R), A83-01 (Alk inhibitor, A) and SB202190 (a p38 MAPK inhibitor, S). For each PDAC organoid line, HPSC media was sequentially depleted of different growth factors (e.g. WENRA, WNRA, ENRA, NRA, ENA, NA, and media alone). If necessary, for example when SB202190 was essential for viability, different combinations of growth factors were depleted. For W deficient media, WRN conditioned was replaced with 5% R-Spondin conditioned media and/or recombinant mouse Noggin (100ng/ml).

**Niche factor depletion studies.** For niche factor depletion studies, PDAC organoid lines initially grown routinely in WENRAS media were collected and digested with TrypLE to generate single cells. They were then filtered through a 40 μm cell strainer. The single cells were replated in Matrigel and grown in basal HPSC media for a minimum of 24 hours to recover cell viability and growth. After 24 hours, single cell/small PDAC organoids were collected, washed, filtered and resuspended in fresh Matrigel (~500–1000 single cell/small organoids per 3ul Matrigel). and the organoids were then plated in white walled 96-well plates. A minimum of 6 replicates were tested for each niche factor media condition and results were compared to control HSPC media. Organoid cultures were fed with 150ul media/well and refed every 2 days. After 7 days of culture, cell viability was assessed by CellTiter-Glo 3D Luminescent Cell Viability Assay. Data was normalized to organoid viability under control HPSC media conditions. Minimal niche factor conditions that supported between 50–100% viability of HPSC media were selected for Wnt inhibitor studies. One-way ANOVA was utilized for statistical analysis.

## Validation of Wnt dependency using Porcupine and Wnt inhibitors

To confirm the Wnt (in)dependency of individual PDAC organoid lines, the effects of ETC-159 (PORCNi), C59 (PORCNi) and ICG-001 (Wnt/β-catenin inhibitor) on organoid viability were examined. Using the minimal niche factor conditions defined in above niche factor depletion studies, each PDAC tumor organoid line was treated with vehicle alone, ETC-159 (1–10μM), C59 (1–10μM) or ICG-001 (1 and 10μM). Briefly, single cell/small PDAC organoids were prepared and plated in Matrigel as described above. A minimum of 6 replicates were tested for each inhibitor condition and results were compared to control minimal niche factor conditions. Organoid cultures were fed with 150μl media/well and refed every 2 days.

After 7 days culture, cell viability was assessed using the CellTiter-Glo 3D Luminescent Cell Viability Assay as described above. Based on the treatment effects to both ETC-159 and C59, Wnt-dependency was defined by a >75% reduction in viability at both doses (1μM and 10μM). Partial Wnt-dependency was defined by 50–75% reduction with at least one inhibitor at the 10μM dose. Wnt (in)dependency was defined by a <50% reduction in viability after PORCNi treatment. One-way ANOVA was utilized for statistical analysis.

## Assessment of Wnt inhibitors as single agents or in combination with standard of care chemotherapy *in vitro*

Panc269 and Panc320 as the selected PDAC tumor organoids in the PTOL were prepared for growth inhibition assays as described above. Prior to replating in the minimal niche factor media (Panc129: ENRA, Panc193: ENRAS, Panc269: R, Panc271: ENRA, Panc308: ENRA, Panc272: media alone, Panc268: A, Panc320: ENA), PDAC tumor organoids were labeled for one hour with Calcein AM live cell dye (ThermoScientific). Twenty-four hours after replating, the PORCNi ETC-159, gemcitabine, paclitaxel, the combination of ETC159/gemcitabine, ETC-159/paclitaxel, or the triple combination were added at a range of concentrations as previously described [30, 31]. After day 7, viability was evaluated in each well using CellTiter-Glo 3D Luminescent Cell Viability assay. The enhanced combination effects were determined by statistical analyses described below.

## Statistical analysis of *in vitro* organoid drug treatment assays

GraphPad Prism software version 9.0 was used to perform statistical analyses. One-way ANOVA with multiple comparisons test was used for all analyses. *P = 0.05* was used as the cutoff for statistical significance. Error bars represent standard error of the mean.

   **Single agent drug treatment assay.** The PORCNi ETC-159 was tested at five dose levels (0.313 μM, 0.625 μM, 1.25 μM, 2.5 μM, 5 μM), paclitaxel was tested at three dose levels (2.5 nM, 5 nM, 10 nM,) and gemcitabine was tested at three dose levels (0.125 μM, 0.25 μM, 0.5 μM). Each of these doses were tested independently as single agents to assess growth and anti-proliferative effect.

   **Two drug combination drug treatment assay.** Each of the drugs listed above (ETC-159, 1μM, paclitaxel 1nM, gemcitabine, 1nM) were utilized in all possible two drug combinations. These compounds were then used to analyze growth and anti-proliferative two drug combinatory effect.

   **Analysis of anti-proliferative effect of *in vitro* drug treatment assay.** The anti-proliferative effects were evaluated by using the Area Under the Curve (AUC) from the plot of organoid growth (amount of fluorescence) over time. This allowed for determining which combinations provide the most promising anti-proliferative effect. Comparisons were also drawn between any linear combination of treatments in these *in vivo* experiments. Experiments were repeated twice for each selected PDAC tumor organoid line. End-point CellTiter Glo 3D assays was used to validate results from AUC over time studies. From the 96-well plate proliferation data, growth curves over time were created.

## Evaluation of the efficacy of Wnt inhibitors as single agents or in combination with standard of care chemotherapy in PDAC PDX models

To confirm the anti-proliferative effects of the above drug combinations on PDAC tumor organoids, we performed similar studies in PDAC PDX models. Based on the subtyping of Wnt (in)dependency in PDAC tumor organoids and drug responses, we selected Panc269

(Wnt autocrine dependent) and Panc320 (Wnt independent) PDX models. Each PDX model was treatedwith compounds as single agents and combinations that demonstrated the best combinatorial effects. The selected PDAC PDX models were implanted subcutaneously into the flanks female athymic nude mice. When the tumors reached approximately 100-300mm$^3$, they were randomized into one of 6 groups: Vehicle, ETC-159 (30mg/kg, PO, QD), gemcitabine (40mg/kg, IP, QW), paclitaxel (30mg/kg IV, for 5 consecutive days), ETC-159/gemcitabine, and ETC-159/paclitaxel. The doses used for combinatory treatment were the same as listed prior. These drug concentrations were chosen based on previous data from our lab with minimal toxicities (weight loss and decreased body condition) seen in the combinations [31, 32]. Panc269 had a total of 47 mice (7–8 mice per group) and Panc320 a total of 38 mice (6–7 mice per group). Mice were monitored daily for health and signs of toxicity, weighed twice per week, and tumors were measured twice weekly using digital calipers using the following formula: tumor volume = (length x width$^2$) x 0.52. Following treatment for 28 days, several mice from each group were euthanized for gene expression and pharmacodynamic analyses. The remaining mice had treatment discontinued and regrowth of tumors for an additional three months or until experimental endpoints or euthanasia criteria was reached. Few toxicities were noted, but no mice were found dead before meeting criteria for euthanasia. Euthanasia criteria per our IACUC protocol included 15% weight loss, one tumor reaching 2000mm$^3$ or total tumors 3000mm$^3$, tumor ulcerations, body condition score of 2 or less, or sickly/moribund mice. When mice reached euthanasia criteria, they were euthanized within 24–48 hours. Mice received daily supplemental food, supportive care when needed, and analgesics for pain or distress with Veterinary staff evaluations.

## Whole exome sequencing analysis

Genomic DNA from organoids and PDX were purified using Zymo Genomic DNA kit (ZymoResearch, Irvine, CA) and sent to Novogene for whole exome sequencing [33]. VCF files with variant calls were generated by the Novogene bioinformatics pipeline. Specifically, sequence reads were trimmed to remove adapters and low quality bases and aligned to the hg38 assembly of the human genome using the Burrows-Wheeler aligner BWA-MEM (v0.7.17). Variants were called and VCF files generated using the GATK HaplotypeCaller (version 4.0.5.1). Further analysis was performed at the University of Colorado Cancer Center Biostatistics and Bioinformatics Shared Resource. VCF2MAF (version 1.6.21) was used to assemble tables with canonical transcripts/proteins from the Novogen VCF files which were then filtered for variants impacting the protein.

## RNA-sequencing analysis

For transcriptome analysis, all PDAC tumor organoids in the PTOL were grown to a sufficient density in complete HPSC media. Total RNA was purified and high-quality RNA for standard transcriptome was analyzed on the Illumina NovaSeq6000. Illumina adapters and reads <50 base pairs were removed using BBDuk (v38.90) [34]. Remaining reads were aligned simultaneously to human (GRCh38) and mouse (GRCm39) genomes using STAR (v2.7.9a) with splice junction information from Ensembl release 104 [35, 36]. Reads mapping unambiguously to the human genome were counted and assigned to genes using feature Counts (v2.0.3) [37]. Ensembl IDs were converted to gene names and the counts of genes with multiple IDs were aggregated. Counts were normalized to counts per million (CPM) and genes with a mean count <1 CPM were discarded from further analysis. The GSVA R package (v1.40.1) was used with default settings to score each sample based on Wnt dependent and independent gene signatures [1, 38]. These scores were z-score transformed. Heatmaps were produced using the Complex Heatmap R package (v2.8.0) [39].

## Gene expression studies through RT-PCR

To gain further insights into the relationship between combination drug responses and Wnt dependency, targeted gene-expression studies were performed based on transcriptional profiling above. A major focus was gene expression changes in Wnt pathway and target genes following different drug treatments. The doses used for each treatment were ETC-159 at 15mg/kg, paclitaxel 10 mg/kg, and gemcitabine 40 mg/kg. Following tumor harvest from mice after treatment as described previously, total RNA was purified and cDNA synthesis was performed. Quantitative RT-PCR with TaqMan Gene Expression Assay for Human Wnt (Thermo Fisher, A6102) using TaqMan primer/probes (Thermo Fisher) were performed on both Wnt pathway and target genes: WNT ligands (e.g. WNT3, WNT3A, WNT7A, WNT7B, WNT10A, etc), WNT receptors (e.g. Frizzled-7/8, etc), R-Spondin signaling (e.g. Lgr5, RSpondin ligands), Wnt inihibitors (e.g. Dickkopf proteins (DKKs), secreted Frizzled related proteins (sFRP) etc), additional Wnt signaling components (e.g. APC, GSK3b), other direct Wnt target genes (e.g. ASCL2, PTPRO, BMP4/7, AXIN2, NKD1/2, ZNRF3, NKD1, SP5, PATZ1, RNF43, APCDD1, NOTUM) and Wnt related transcription factors (e.g. GATA6, FOXA2).

## Results

### Evaluation of whole exome sequencing in PDAC organoid lines

Mutational analysis from our original patient tumors revealed several notable key driver mutations as previously identified. Given this clinical mutation data, whole exome sequencing was utilized to evaluate the genetic mutational profiles of the PDAC organoids used in this study. Major contributors to the development of pancreatic cancer were of particular emphasis, namely TP53, KRAS, CDKN2a, and SMAD4 [2, 3]. We checked the sequences generated from Panc268, Panc269, Panc320, and respective organoids for mutations in these genes. Table 1 shows the pathogenic mutations identified in accordance with ClinVar [40].

Clinical mutational profiling of PTOL tumors revealed the same frequency of oncogene (e.g. KRAS) and tumor suppressor gene (e.g. TP53) mutations as the general pancreatic cancer population (Table 1). For example, Panc269 expresses common KRAS and TP53 gene mutations whereas Panc272 has no detectable gene mutations. This did not correlate with niche factor dependency determined below.

### Assessment of niche factor dependency in PDAC organoid lines

Eight PDAC organoid lines from our PTOL were tested for minimal niche factor requirements. As shown in Fig 1, there were distinct growth factor dependencies among the PDAC organoid lines. Panc129, Panc271 and Panc308 readily grew independently of exogenous Wnt but required exogenous R-spondin for viability. Panc193 showed a similar niche factor dependency, but it was also dependent on the presence of the p38 MAPK inhibitor SB202190. On the other hand, Panc268 and Panc269 were Wnt-independent with different single niche factor conditions. Panc269 grew independently of exogenous Wnt and other niche factors (E, N, A and S) but had a strict requirement for exogenous R-Spondin, whereas Panc268 showed strong viability in N alone, R-Spondin alone or A83-01 alone media conditions. In contrast, Panc320 and Panc272 grew in the absence of both exogenous Wnt and R-Spondin. Viability of Panc320 still required the presence of Noggin and A83-01, whereas Panc272 grew independently of all exogenous niche factors (Fig 1). A summary of Wnt/R-spondin dependency in the pancreatic organoids characterized through growth factor dependency and Wnt pathway inhibition is displayed in S1 Table.

**Table 1. Mutations of key players in Wnt pathway identified through whole exome sequencing of respective organoids, PDX PDAC, and clinical mutations data from patients [31].**

| Sample | Gene | Mutation | Affect |
|---|---|---|---|
| Panc269 | CDKN2a | A148T | benign |
| | KRAS | Q61H | pathogenic |
| | SMAD4 | G255A | US* |
| | TP53 | S127F | conflicting interpretations of pathogenicity |
| | | P72R | benign |
| OPanc269 | CDKN2a | A148T | benign |
| | KRAS | Q61H | pathogenic |
| | TP53 | S127F | conflicting interpretations of pathogenicity |
| | | P72R | benign |
| Panc269—patient | CDKN2a | E26_L32delinsV | US* |
| | KRAS | Q61H | pathogenic |
| | TP53 | S127F | conflicting interpretations of pathogenicity |
| Panc320 | CDKN2a | L32P | pathogenic |
| | SMAD4 | G230A | conflicting interpretations of pathogenicity |
| | | G255A | US* |
| | | I347V | US |
| | TP53 | R282W | pathogenic |
| | | P72R | benign |
| OPanc320 | CDKN2a | L32P | pathogenic |
| | KRAS | G12D | pathogenic |
| | TP53 | R282W | pathogenic |
| | | P72R | benign |
| Panc320—patient | CDKN2a | L32P | Pathogenic/likely pathogenic |
| | KRAS | G12D | pathogenic |
| | SMAD4 | D415E* | US* |
| | TP53 | R282W | Pathogenic/likely pathogenic |

US* = no reported data. US = unspecified significance. Delins = deletion-insertion Letters and numbers demonstrate specific amino acid substitutions.

## Evaluation of Wnt (in)dependency using Porcupine inhibitor in PTOLs

Previous work from the Sato lab had examined different Wnt (in)dependencies in PDAC organoid lines using Porcupine inhibitors [1]. To further investigate the requirement of endogenous Wnt secretion on PDAC viability, each PDAC organoid line was grown in minimal niche factor media that supported >50% viability as defined in Fig 1 and treated with 1uM and 10uM of the PORCNi ETC-159 and C59 (Fig 2). Treatment with the two different PORCNi (ETC-159 and C59) resulted in a strong inhibition of growth (>75% decrease) in four PDAC organoid lines (Panc193, Panc269, Panc271 and Panc272). These organoids have an essential requirement for endogenous Wnt secretion for viability. Of note, Panc269 was a striking example of a PDAC organoid that required endogenous Wnt secretion and paracrine R-Spondin for growth. Partial PORCNi responses (50%-75% decrease) were observed with Panc129 and Panc308. In contrast, Panc272, which grew independently of all niche factors including R-Spondin, was still dependent of endogenous Wnt secretion for its growth. The two other PDAC organoid lines, Panc268 and Panc320 were resistant to PORCNi treatment (<50% decrease) indicating that organoid growth was independent of Wnt secretion (Fig 2).

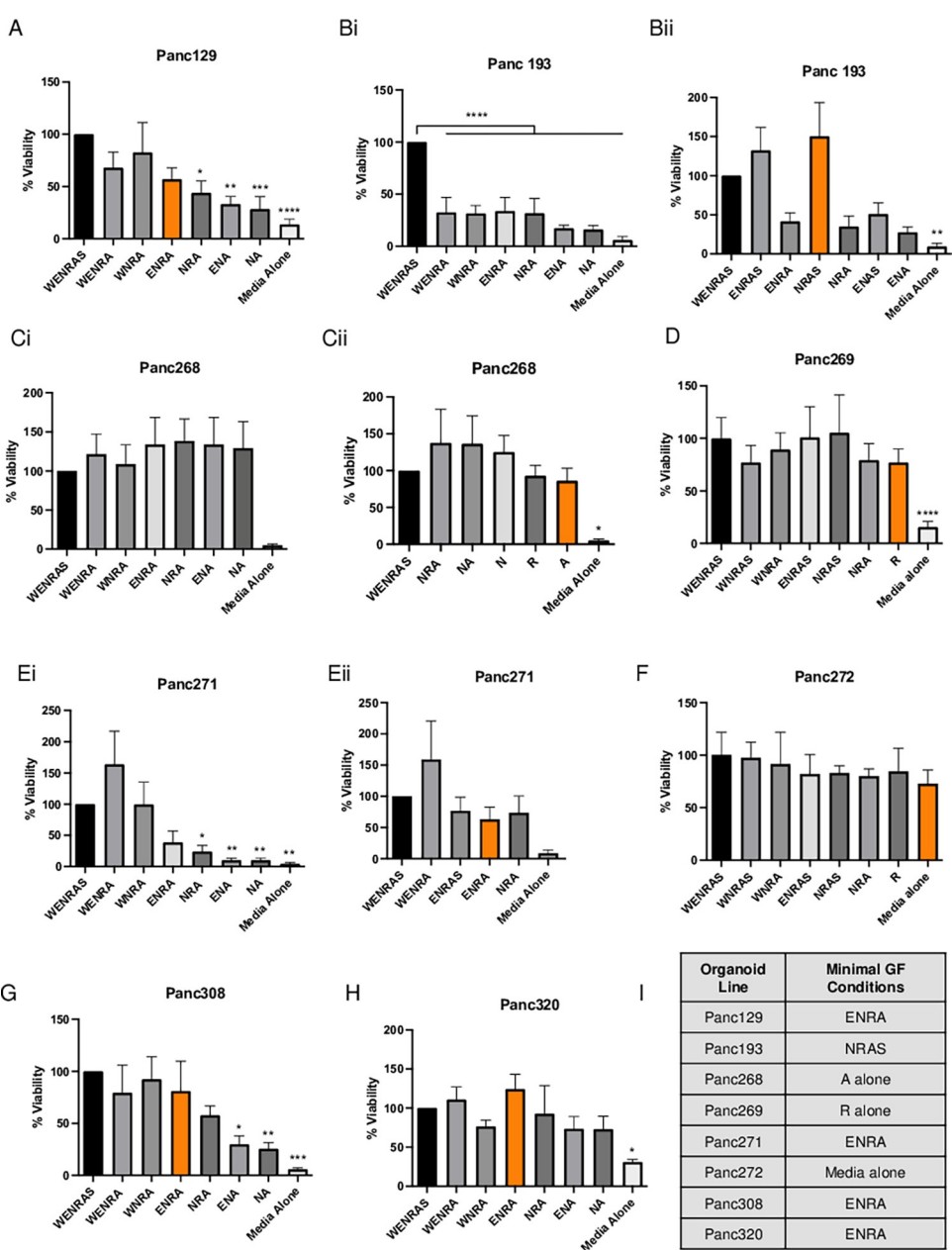

**Fig 1. Minimal growth factor conditions for PTOL.** All growth is expressed as percent viability respective to control (complete WENRAS media), with minimal growth factor media required to supported >50% viability assessed for each organoid. Minimal conditions highlighted for each as applicable, including ENRA for Panc129 (A), NRAS for Panc193 (Bii), A for Panc268 (Cii), R for Panc269 (D), ENRA for Panc271 (Eii), media without supplemented growth factor for Panc 272 (F), ENRA for Panc308 (G) and Panc320 (H). These are additionally highlighted in table form (I). The organoids without minimal growth factor conditions highlighted required all growth factors (WENRAS) for growth (Bi, Ci, Ei). W = 50% Wnt3a. E = 50 ng/ml EGF. N = Noggin. R = R-spondin. A = 500 nM A38-01 (ALK inhibitor). S = 10 uM SB202190 (p38 MAPK inhibitor). Statistical analysis done via one-way ANOVA.

## Examination of (in)dependency utilizing Wnt gene signatures in pancreatic organoids and PDX tumors

To assess whether Wnt (in)dependency phenotypes could be validated at the gene expression level, each pancreatic organoid and PDX tumor were scored based on Wnt dependency using

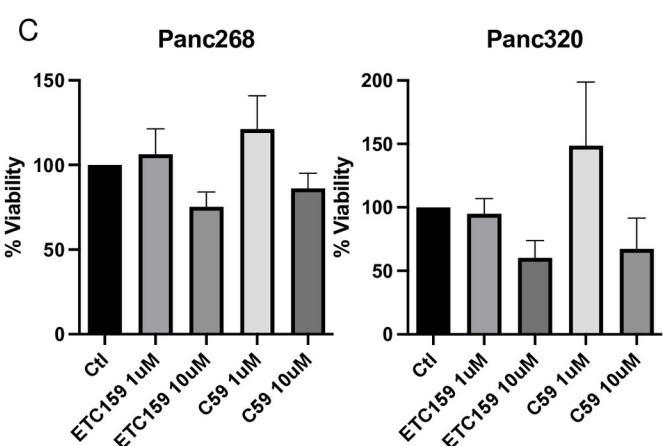

**Fig 2. Wnt dependency confirmed through the utilization of Wnt pathway inhibitors: ETC159, C59.** All growth is expressed as percent viability respective to control (no drug). For all Wnt pathway inhibition assays, organoids were grown in minimal growth media (Panc129: ENRA, Panc193: ENRAS, Panc269: R, Panc271: NRA, Panc308: ENRA, Panc272: media alone, Panc268: A, Panc320: ENA). Each was then treated with either 1uM or 10uM of Wnt inhibitor, ETC-159 and C59 were utilized in this study. W = 50% Wnt3a. E = 50 ng/ml EGF. N = Noggin. R = R-spondin. A = 500 nM A38-01 (ALK inhibitor). S = 10 uM SB202190 (p53 MAPK inhibitor). ND = no drug. Ctl: = control (no drug). Statistical analysis was done via one-way ANOVA.

a gene signature previously described by Sato's group (Fig 3 and S1 Fig and S1 Table and S2 Table) [1]. As it pertains to the pancreatic organoids that underwent combinatory drug treatment and investigation *in vivo*, Panc269 demonstrated a gene signature consistent with Wnt dependency, and Panc268 demonstrated a Wnt independent gene signature. Of note, Panc320 that was also defined as a Wnt independent model, demonstrated a more neutral gene signature despite demonstrating Wnt independent signaling (Fig 3).

## Drug treatment of PDAC organoids and xenografts

Panc269 (Wnt dependent) and Panc320 (Wnt independent) PDAC organoids were selected to evaluate the treatment effects of the WNT inhibitor ETC-159 in combination with gemcitiabine or paclitaxel as known standards of treatment of PDAC [17]. When treated with single agent alone (ETC-159, paclitaxel or gemcitabine), ETC-159 had a greater decrease in proliferation when compared to control as well as with paclitaxel and gemcitabine alone in the Panc269 organoid (Fig 4A). There were no differences in treatment effects when the chemotherapeutic agents paclitaxel and gemcitabine were added to ETC-159 as proliferation in the combination groups were similar to that of ETC-159. In contrast, there were no single agent ETC-159 or combination effects with respect to proliferation in the Panc320 PDO model (Fig 4B).

To confirm the *in vitro* treatments effects in our *in vivo* PDAC explant mouse model, Panc269 and Panc320 were treated with ETC-159 alone and in combination with gemcitabine or paclitaxel. As displayed in Fig 4C, there was a slight decrease in the tumor growth rate with single agent treatment ETC-159 when compared to vehicle in Panc269, however this was not statistically significant. In addition, no combination treatment effects were identified in Panc269 when the agents (paclitaxel and gemcitabine) were added to ETC-159. In Panc320, there were no single agent or combinational treatment effects observed in this explant (Fig 4D). Tumor regrowth was minimal for both Panc269 and Panc320 following the removal of drug (S2 Fig).

## Human Wnt pathway RT-PCR assay of post-treatment pancreatic tumors

In order to further elucidate the mechanism of Wnt (in)dependency, treatment sensitivities, and treatment resistance in the pancreatic organoids used in this study, we performed RT-PCR on each tumor after treatment with ETC-159, paclitaxel, gemcitabine, and ETC-159 in combination with each respective chemotherapy. In a prior study, AXIN2, CTNNB1, C-MYC, MKD1, and TCF7 demonstrated decrease in expression following treatment with ETC-159 [27]. Panc269 demonstrated a decrease in NKD1, MYC and Axin2 when treated with ETC-159 when compared to vehicle (Fig 5A). This was not the case in Panc320, where the genes were all upregulated in the ETC-159 group when compared to vehicle (Fig 5B). Additionally, we looked at expression of various other genes pertaining to the Wnt signaling pathway by calculating fold change expression following respective treatment over vehicle (Fig 5C). Of importance, Panc320 demonstrated an increase expression in FZD7 and PYGO2, whereas Panc269 showed an increase expression in FRZB and WIF1.

## Discussion

Pancreatic cancer is known to be a leading cause of cancer death due to its presentation at an advanced stage and resistance to current chemotherapeutic treatment regimens [41]. A key contributor to the development and resistance of PDAC may be the reactivation of Wnt/R-Spondin signaling pathway [2, 3]. Our *in vitro* studies classifying PDAC organoids according to Wnt (in)dependency confirmed the significance of this signaling pathway, consistent with previous studies done by Sato's group [1]. These data suggest that targeting of the Wnt

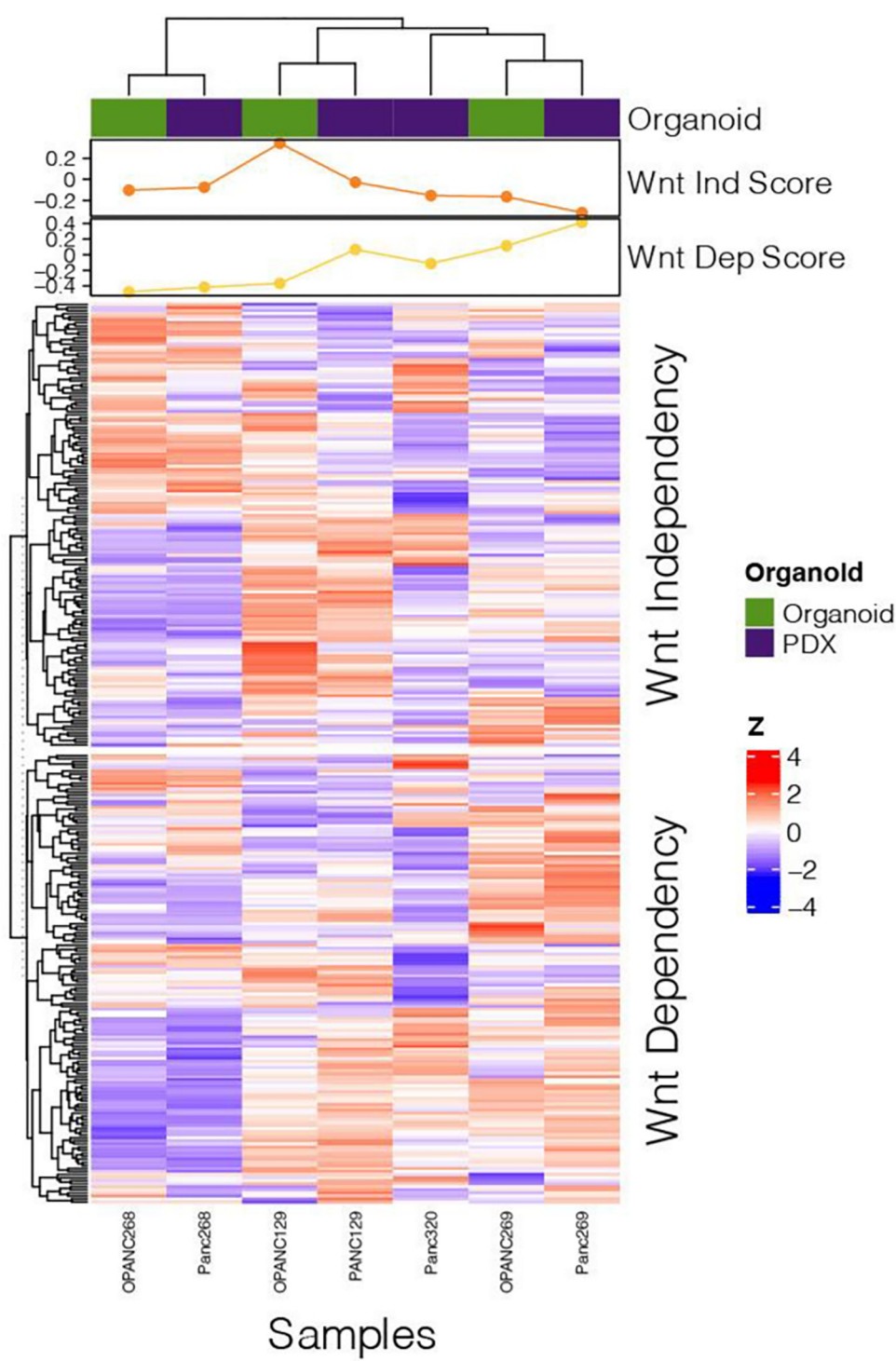

**Fig 3. Wnt (in)dependency gene signatures of select Panc samples according to those identified by Seino et al. [1].**
Heatmap displaying Wnt (in)dependent gene signatures for PDX PDAC tumor and organoid lines. Gene names as described by Seino et al referenced in S2 Table. The dependency scores are generated using GSVA. The CPM of each gene was z-score transformed across each row. Panc268, Panc269, and Panc320 organoid lines compared to their respective PDX tumors. Wnt independent gene signature of Panc268 and Wnt dependent gene signature of Panc269 based on z-score transformation consistent with growth factor dependency data. OPANC = organoid, PANC = PDX tumor.

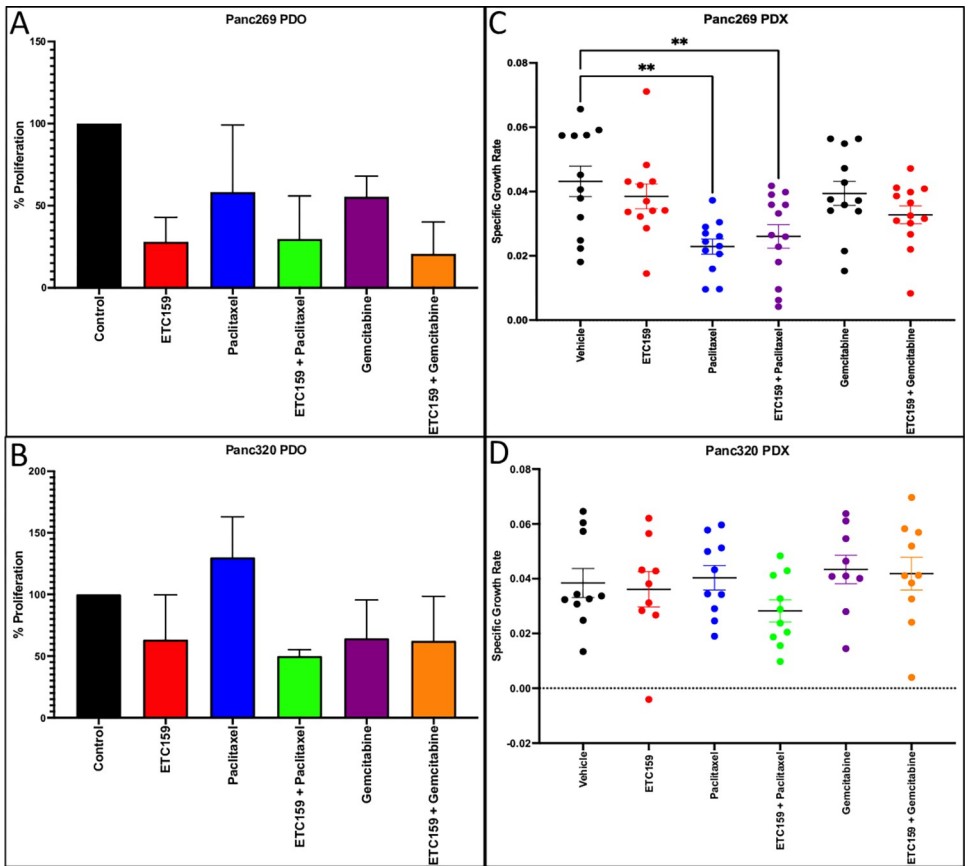

**Fig 4.** *In Vivo* **combinatory drug treatment.** A) PDAC patient-derived organoids treated with single agent and combination treatment. B) Athymic nude mice with subcutaneously injected tumors were treated for 28–31 days with single agent or combination of ETC-159 and either Paclitaxel or Gemcitabine. Specific growth rate was calculated, demonstrating more affective reduction of growth rate with combinatory treatment of Wnt dependent Panc269, consistent with *in vitro* data. Panc320 also demonstrated significant reduction in growth rate with combination of ETC-159 and Paclitaxel, but otherwise growth was not as affected in this Wnt independent model supported by *in vitro* data.

pathway through inhibitors (i.e. ETC-159) could be used in the treatment of pancreatic cancer, especially those with Wnt dependent phenotype. We did confirm effectiveness of Wnt inhibition through the utilization of ETC-159 and C59, with a decrease in proliferation of Wnt dependent tumors. A prior study done in 2016 by Madan et al. demonstrated decreased expression of AXIN2, C-MYC, NKD1, and TCF7 following treatment of pancreatic tumors with ETC-159. These data were confirmed by our RT-PCR assay evaluating Wnt signaling pathway gene expression. There was decreased expression following ETC-159 treatment in our Wnt dependent Panc269 organoid compared with our Wnt independent Panc320 organoid. The increased expression of these genes observed in Panc320 suggests a potential mechanism of resistance [30]. Additionally, we observed a trend of decreased tumor growth in our mouse models injected with the Wnt dependent tumors relative to the mice injected with Wnt independent tumors when treated in combination with gemcitabine or paclitaxel and ETC-159.

A major aspect of the clinical application of this novel therapeutic approach of pancreatic cancer is the utilization of ETC-159 in combination with the current standard chemotherapies, nab-paclitaxel and gemcitabine [17–19, 25]. There was trend for a decrease in proliferation of

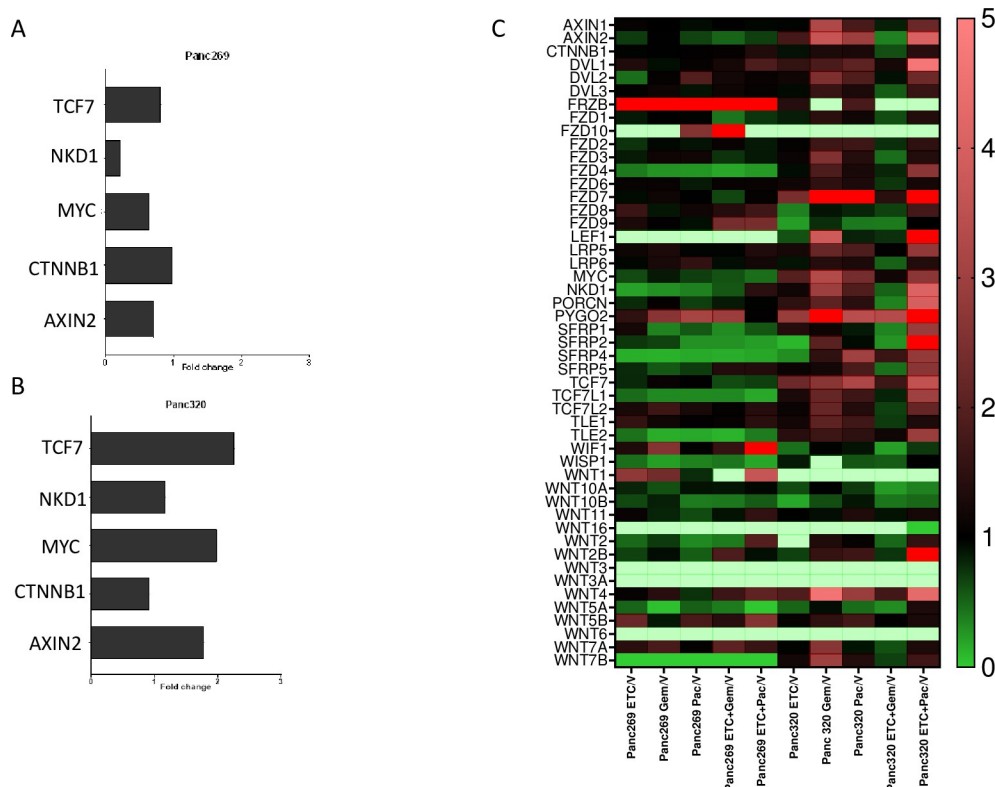

**Fig 5. RT-PCR gene expression analysis of organoids post-treatment.** RT-PCR was performed on post-treatment tumors of Panc269 (A) and Panc320 (B) harvested from mice looking at gene expression of genes previously published by Madan et. al. that are known to decrease following treatment with ETC-159. C) Fold change gene expression following treatment with ETC-159 over vehicle was calculated and displayed as heat map. These previously published data are consistent with Panc269, though these genes demonstrated increased expression in Panc320 following treatment with ETC-159.

our Wnt dependent PDAC organoids with combinatory treatment relative to monotherapy with any agent. This is suggestive of a more effective reduction in tumor burden for patients treated with standard chemotherapy and ETC-159, with a potential for more successful surgical debulking or palliative symptom reduction [20]. This was not as evident in the xenograft models. However, Panc320 did show a modest benefit with the combination of ETC-159/paclitaxel when compared to either single agent. While this was observed in the setting of Wnt dependency, Wnt independent tumors would need further mechanistic evaluation as Wnt inhibition does not appear to be as effective *in vivo*. This may be due to lack of essential growth factors in the mouse microenvironment.

In addition to resistance to ETC-159 seen in Panc320, certain gene expression changes have been identified in chemotherapeutic resistance. These may explain the lesser growth rate reduction seen *in vivo* following combinatory treatment with ETC-159 and chemotherapy. FRZB expression was notably increased in Panc269, which has been found to progress pancreatic cancer through non-canonical Wnt signaling pathway [42]. This could provide perspective as to one avenue of Wnt dependency and mechanism of cancer progression when overexpressed. PYGO2 expression was increased in Panc320, providing evidence of mechanism of resistance to Wnt inhibition and chemotherapy. When inhibited, this gene has been shown to restore chemotherapeutic drug sensitivity in breast cancer stem cells, which may also explain chemotherapeutic resistance observed in the organoids used in this study [43]. FZD7/WNT7b

signaling have been identified as contributors to chemoresistance in pancreatic cancer [44]. These genes were both elevated in Panc320 following treatment with ETC-159 and paclitaxel and may explain the persistent growth rate. In Panc269, FZD7 expression is decreased following treatment and more sensitive to combinatory treatment as evident by decreased growth rate. SFRP2 was also elevated in Panc320 but decreased in Panc269 following treatment with chemotherapy, or in combination with ETC-159. This could also lead to chemo-resistance through Wnt16 when upregulated [45]. WIF1 was upregulated in Panc269, especially following combinatory treatment with ETC-159 and Paclitaxel. It has been previously shown that overexpression of WIF1 increased the efficacy of paclitaxel, providing explanation as to why Panc269 appeared to be more sensitive to ETC-159/Paclitaxel treatment when compared with ETC-159/Gemcitabine [46]. While further genetic profiling would need to be completed to better define the mechanism behind the different treatment sensitivities observed in various pancreatic tumors, these data would be reasonable explanations as to the decrease in growth rate observed *in vivo* following combinatory treatment with ETC-159 and chemotherapy resulting from gene expression changes seen following treatment.

As emphasized by this study, the genotype and biologic phenotypes are difficult to associate to allow for accurate correlation [6]. While we have adapted the utilization of organoids to help bridge this gap between genotype and phenotype [29], this gap remains a challenge when applying *in vitro* data to a more clinical context. We were able to effectively classify the phenotypes of our PDAC organoids in light of Wnt signaling and response to inhibition and chemotherapy. The question of relation to genotype remained a key question when considering the mechanism behind our data. In addition to the PCR data obtained, heatmapping of Wnt dependent and Wnt independent genes confirmed Wnt independent gene signatures for most Wnt independent organoids, and Wnt dependent gene signatures for most Wnt dependent organoids, again consistent with previous work done by Sato's group [1]. Consistent gene signatures that could be used to more efficiently classify Wnt (in)dependency in pancreatic tumors. This classification of gene signatures of PDAC PTOLs could be utilized as a prognostic indicator for consideration of Wnt inhibition therapy in combination with standard chemotherapy for those tumors dependent on Wnt signaling for growth. It could additionally serve as a tool to better predict expected phenotypes for patient derived pancreatic organoids that are developed. Furthermore, we need to validate the differences seen between the organoid and its complimentary PDX. This may include introducing various growth factors to the mouse to compensate for the differences seen in the tumor microenvironment vs the growth media for the organoid.

This study was largely limited by the size of our PDAC organoid library and the utility of these organoids dictated by varying growth rates. For more universal acceptance of results, these experiments will need to be conducted on more organoids as they are developed.

## Supporting information

**S1 Fig. Wnt (in)dependency gene signatures of all Panc samples according to those identified by Seino et al. [1].** Heatmap displaying Wnt (in)dependent gene signatures for all PDX PDAC tumor and organoid lines. Gene names as describe by Seino et al referenced in S2 Table. The dependency scores are generated using GSVA. The CPM of each gene was z-score transformed across each row. Pancreatic organoid lines compared to their respective PDX tumors. OPANC = organoid, PANC = PDX tumor.
(TIF)

**S2 Fig. *In Vivo* combinatory drug treatment.** Athymic nude mice with subcutaneously injected tumors were treated for 30 days with single agent or combination of ETC-159 and

either Paclitaxel or Gemcitabine. Specific growth rate was calculated, demonstrating more affective reduction of growth rate with combinatory treatment of Wnt dependent Panc269, consistent with *in vitro* data. Panc320 also demonstrated significant reduction in growth rate with combination of ETC-159 and Paclitaxel, but otherwise growth was not as affected in this Wnt independent model supported by *in vitro* data.
(TIF)

**S1 Table. Summary of pancreatic organoids Wnt/R-spondin dependency established through growth factor dependency and Wnt pathway inhibition.**
(TIF)

**S2 Table. Order of genes included in heat map of Wnt (in)dependency gene signatures of select Panc samples demonstrated in Fig 3 and S2 Fig.**
(XLSX)

## Acknowledgments

This study was conducted under the support of Todd Pitts and Peter Dempsey as Co-PIs.

## Author Contributions

**Conceptualization:** Hayley J. Hawkins, John J. Arcaroli, Stacey M. Bagby, Robert W. Lentz, Christopher H. Lieu, Alexis D. Leal, Wells A. Messersmith, Peter J. Dempsey, Todd M. Pitts.

**Data curation:** Hayley J. Hawkins, Betelehem W. Yacob, Monica E. Brown, Brandon R. Goldstein, John J. Arcaroli, Stacey M. Bagby, Sarah J. Hartman, Andrew Goodspeed, Thomas Danhorn, Peter J. Dempsey, Todd M. Pitts.

**Formal analysis:** Hayley J. Hawkins, Stacey M. Bagby, Andrew Goodspeed, Thomas Danhorn, Christopher H. Lieu, Peter J. Dempsey, Todd M. Pitts.

**Funding acquisition:** Todd M. Pitts.

**Investigation:** Hayley J. Hawkins, Betelehem W. Yacob, John J. Arcaroli, Stacey M. Bagby, Sarah J. Hartman, Morgan Macbeth, Peter J. Dempsey, Todd M. Pitts.

**Methodology:** Hayley J. Hawkins, Betelehem W. Yacob, Monica E. Brown, John J. Arcaroli, Stacey M. Bagby, Sarah J. Hartman, Andrew Goodspeed, Peter J. Dempsey, Todd M. Pitts.

**Project administration:** Todd M. Pitts.

**Resources:** Sarah J. Hartman, Robert W. Lentz, Alexis D. Leal, Peter J. Dempsey, Todd M. Pitts.

**Supervision:** Monica E. Brown, Sarah J. Hartman, Peter J. Dempsey, Todd M. Pitts.

**Validation:** Hayley J. Hawkins, Brandon R. Goldstein, Peter J. Dempsey, Todd M. Pitts.

**Visualization:** Hayley J. Hawkins, Peter J. Dempsey, Todd M. Pitts.

**Writing – original draft:** Hayley J. Hawkins.

**Writing – review & editing:** Hayley J. Hawkins, John J. Arcaroli, Andrew Goodspeed, Peter J. Dempsey, Todd M. Pitts.

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
