## [Decision Letter · Decision Letter 0]

26 Dec 2023

PONE-D-23-36221Examination of Wnt signaling as a therapeutic target for pancreatic ductal adenocarcinoma (PDAC) using a pancreatic tumor organoid library (PTOL).PLOS ONE

Dear Dr. Hawkins,

Thank you for submitting your manuscript to PLOS ONE. After careful consideration, we feel that it has merit but does not fully meet PLOS ONE’s publication criteria as it currently stands. Therefore, we invite you to submit a revised version of the manuscript that addresses the points raised during the review process. Please submit your revised manuscript by Feb 09 2024 11:59PM. If you will need more time than this to complete your revisions, please reply to this message or contact the journal office at plosone@plos.org. Please include the following items when submitting your revised manuscript:A rebuttal letter that responds to each point raised by the academic editor and reviewer(s). You should upload this letter as a separate file labeled 'Response to Reviewers'.A marked-up copy of your manuscript that highlights changes made to the original version. You should upload this as a separate file labeled 'Revised Manuscript with Track Changes'.An unmarked version of your revised paper without tracked changes. You should upload this as a separate file labeled 'Manuscript'.

We look forward to receiving your revised manuscript.

Kind regards,

Yash Gupta, Ph.D.

Academic Editor

PLOS ONE

Journal Requirements:

"This study was funded by The Wings of Hope for Pancreatic Cancer Research Pilot Grant"

"This study was funded by The Wings of Hope for Pancreatic Cancer Research Pilot Grant, with Todd Pitts, and Peter Dempsey as Co-PIs. This study was partly supported by the National Institutes of Health P30CA046934 Bioinformatics and Biostatistics Shared Resource and Organoid and Tissue Modeling Shared Resource. "

"This study was funded by The Wings of Hope for Pancreatic Cancer Research Pilot Grant"

5. We note that your Data Availability Statement is currently as follows: [All relevant data are within the manuscript and its Supporting Information files.]

7. We note that you have referenced (unpublished data) on page 9, which has currently not yet been accepted for publication. Please remove this from your References and amend this to state in the body of your manuscript: (ie “Bewick et al. [Unpublished]”) as detailed online in our guide for authors

Additional Editor Comments:

Reviewers have raised critical concerns regarding experiment design, result interpretations and statistical analysis. Authors need to point wise address the expert comments. Statistics may need major revisions.

Reviewers' comments:

Reviewer's Responses to Questions

**Comments to the Author**

1. Is the manuscript technically sound, and do the data support the conclusions?

Reviewer #1: Partly

Reviewer #2: Yes

2. Has the statistical analysis been performed appropriately and rigorously? 

Reviewer #1: No

Reviewer #2: Yes

3. Have the authors made all data underlying the findings in their manuscript fully available?

Reviewer #1: No

Reviewer #2: Yes

4. Is the manuscript presented in an intelligible fashion and written in standard English?

Reviewer #1: Yes

Reviewer #2: Yes

5. Review Comments to the Author

Reviewer #1: The manuscript ‘Examination of Wnt signaling as a therapeutic target for pancreatic ductal adenocarcinoma (PDAC) using a pancreatic tumor organoid library (PTOL)’ by Hawkins et al, studied the Wnt dependent and independent phenotypes of pancreatic tumors using organoids. This reviewer provides the following comments/suggestions to be addressed by the Authors.

1. Did the authors observed any correlation between the mutations identified in the organoids and niche factor dependency? Please explain in detail regarding the same in the manuscript.

2. No much difference in viability was observed between 1μM or 10 μM concentration of ETC-159 and C59. And for combination analysis, ETC-159 was used at concentrations 0.313 μM, 0.625 μM, 1.25 μM, 2.5 μM, 5 μM. Since there is no much effect on viability beyond 1 μM concentration, authors should consider using lower concentrations and a maximum of 1 μM. The concept of analyzing combinatorial drug concentrations is to check if we could decrease the drug concentration and increase the efficacy of treatment by using two different mode of actions. Authors have not mentioned at what combination dose the tumor growth rate was determined or for the gene expression analysis. Figure 4D is missing. On what basis authors have chosen paclitaxel and gemcitabine for combination with ETC-159? May be these two drugs do not have synergistic effect with ETC-159 or the doses used for combination are not appropriate.

3. If combination of paclitaxel and gemcitabine is already known and reported, then authors should have focused on identifying if ETC-159 and combination of paclitaxel and gemcitabine can have increased efficacy. Authors can make an attempt of this combination.

4. Colony formation can be represented as images

5. Authors should carefully go through the figures and tables provided, the supplement figure 2 contains supplement table 2.

6. Statistical analysis for all the graphs needs to be provided

Reviewer #2: Hayley J Hawkins et al

Here are some constructive comments and observations you might consider for the provided based on the provided information about the research involving Examination of Wnt signaling as a therapeutic target for pancreatic ductal adenocarcinoma (PDAC) using a pancreatic tumor organoid library (PTOL), here are some questions you might consider:

I. The text provides a detailed and structured overview of the methods and procedures employed in the study.

II. The division into sections like "Human Specimens," "PDAC Patient-Derived Xenografts," and "Pancreatic Tumor Organoid Library (PTOL) Generation and Characterization" helps organize the information logically.

III. The process of generating PDAC organoid lines is clearly outlined, including details on tissue preparation, digestion, culture conditions, and media components.

IV. Assessment of Wnt Dependency:

a. the methodology for assessing the Wnt (in)dependency of PDAC tumor organoids is thoroughly explained, involving the manipulation of HPSC media and testing the effects on viability.

V. Wnt Inhibitor Studies:

a. The steps involving the use of Porcupine and Wnt inhibitors to confirm Wnt (in)dependency are well-documented, including the criteria used to define dependency.

VI. Minor Comments:

a. In certain sections, consider breaking down longer sentences for enhanced readability.

b. For instance, consider using bullet points or subheadings for better organization in sections with multiple steps.

c. The section on "Statistical Analysis of In vitro organoid drug treatment assays" is comprehensive but complex. Consider breaking down the information into subsections for clarity, separating details about individual drugs and combinations.

d. Clarify the notation "SMAD4(2, 3)." If these numbers represent specific mutations or isoforms, provide a brief explanation. Page 12 results section.

e. For better clarity, consider breaking the long sentence into two sentences: "Viability of Panc320 still required the presence of Noggin and A83-01. In contrast, Panc272 grew independently of all exogenous niche factors." Page 12

f. Clarify the significance or implications of the observed changes in gene expression in Panc320 and Panc269.

6. PLOS authors have the option to publish the peer review history of their article (what does this mean?). If published, this will include your full peer review and any attached files.

Reviewer #1: **Yes: **Priya Arumugam

Reviewer #2: **Yes: **Manish Shukla

---

## [Author Response · Author response to Decision Letter 0]

22 Jan 2024

Journal Requirements:

a. Thank you for the comment regarding formatting requirements. We have used the provided templates, and change the formatting to better fit with the requirements.

a. Thank you for identifying this difference. The funding information section has been updated to accurately match the financial disclosure as “The Wings of Hope for Pancreatic Cancer Research Pilot Grant.” The award number was Wings.2019.004. We have additionally added that the study was partly supported by the National Institutes of Health P30CA046934 Bioinformatics and Biostatistics Shared Resource and Organoid and Tissue Modeling Shared Resource (support grant awarded to the University of Colorado Cancer Center. Please see point #4 for additional funding information and updated statement, as these will both need to be added to financial disclosure statement.

"This study was funded by The Wings of Hope for Pancreatic Cancer Research Pilot Grant"

a. We did not include the role the funders took in the study, and thank the journal for identifying this. The funders had no role in study design, data collection and analysis, decision to publish, or preparation of the manuscript.

"This study was funded by The Wings of Hope for Pancreatic Cancer Research Pilot Grant, with Todd Pitts, and Peter Dempsey as Co-PIs. This study was partly supported by the National Institutes of Health P30CA046934 Bioinformatics and Biostatistics Shared Resource and Organoid and Tissue Modeling Shared Resource. "

"This study was funded by The Wings of Hope for Pancreatic Cancer Research Pilot Grant"

a. Thank you for clarifying that funding information is not to be included in the Acknowledgements Section. The funding sources have been removed from the text. The funding statement needs to be adjusted to state “This study was funded by the Wings of Hope for Pancreatic Cancer Research Pilot Grant (grant number of Wings.2019.004), as well as being partly supported by the National Institutes of Health P30CA046934 Bioinformatics and Biostatistics Shared Resource and Organoid and Tissue Modeling Shared Resource (support grant awarded to the University of Colorado Cancer Center.” Thank you for changing this on my behalf.

5. We note that your Data Availability Statement is currently as follows: [All relevant data are within the manuscript and its Supporting Information files.]

a. Thank you for confirming the data availability statement, and for providing further information regarding the best format for this to be done. I have adjusted the data availability statement to specify that all raw data is available, and have a figshare with DOI of 10.6084/m9.figshare.25035821 that can be used to access the data in a way that is in accordance with the guidelines provided.

a. Thank you for identifying areas that we have included phrases of “data not shown”. These statements have been removed from the manuscript, and the references cited in conjugation with this has been included.

7. We note that you have referenced (unpublished data) on page 9, which has currently not yet been accepted for publication. Please remove this from your References and amend this to state in the body of your manuscript: (ie “Bewick et al. [Unpublished]”) as detailed online in our guide for authors

a. Again, thank you for pointing out where we have referenced incorrectly, and of data that is not yet published or shown. This reference on page 9 has been removed.

Additional Editor Comments:

Reviewers have raised critical concerns regarding experiment design, result interpretations and statistical analysis. Authors need to point wise address the expert comments. Statistics may need major revisions.

a. We thank you for this emphasis regarding the points made above. We have thoughtfully addressed the comments provided and made appropriate revisions. These have been explained in point below in the response to each individual reviewer.

Reviewer 1

The manuscript ‘Examination of Wnt signaling as a therapeutic target for pancreatic ductal adenocarcinoma (PDAC) using a pancreatic tumor organoid library (PTOL)’ by Hawkins et al, studied the Wnt dependent and independent phenotypes of pancreatic tumors using organoids. This reviewer provides the following comments/suggestions to be addressed by the Authors. 

1. Did the authors observed any correlation between the mutations identified in the organoids and niche factor dependency? Please explain in detail regarding the same in the manuscript. 

a. We thank the reviewer for this comment and in recognizing another way in which these mutations could be seen phenotypically. We have reviewed the mutation list associated with each organoid, and correlated such mutations with the niche factor dependency. There was no correlation with mutations identified in Table 1 and niche factor dependency. A sentence was added stated this on page 13 under the section “Evaluation of whole exome sequencing in PDAC organoid lines.”

2. No much difference in viability was observed between 1μM or 10 μM concentration of ETC-159 and C59. And for combination analysis, ETC-159 was used at concentrations 0.313 μM, 0.625 μM, 1.25 μM, 2.5 μM, 5 μM. Since there is no much effect on viability beyond 1 μM concentration, authors should consider using lower concentrations and a maximum of 1 μM. The concept of analyzing combinatorial drug concentrations is to check if we could decrease the drug concentration and increase the efficacy of treatment by using two different mode of actions. Authors have not mentioned at what combination dose the tumor growth rate was determined or for the gene expression analysis. Figure 4D is missing. On what basis authors have chosen paclitaxel and gemcitabine for combination with ETC-159? May be these two drugs do not have synergistic effect with ETC-159 or the doses used for combination are not appropriate. 

a. We thank the reviewer for this comment. This is in regards to the dosing selection used in this study for ETC-159 and C59. There was a significant effect between these doses. The authors in previous work had done dose assessments on multiple colorectal cell lines, and found that there was no significant effect in various doses less than 1μM. Additionally, while most previously published manuscripts use mid-high nM, these are for cell lines and PDOs usually need higher concentration to achieve any response. Therefore, we decided in our initial study design to use these two doses to evaluate for growth inhibition. We have provided these data as well. 

b. The combination dose used for the tumor growth rate and gene expression analysis have been included into the methods. The titled sections these modifications were made were “Evaluation of the efficacy of Wnt inhibitors as single agents or in combination with standard of care chemotherapy in PDAC PDX models” on pages 9-10 and “gene expression studies through RT-PCR” on page 11.

c. We have also made sure that all figures mentioned in the manuscript have clear and accurate corresponding figured attached, as well any corrections to references made within the text.

d. In regards to the basis of selection of selection of paclitaxel and gemcitabine for combination with ETC-159, they are the standard of care for treatment of pancreatic cancer. This has been clarified in the manuscript and can be found in the results section titled “Evaluation of the efficacy of Wnt inhibitors as single agents or in combination with standard of care chemotherapy in PDAC PDX models” on page 9 as well as “Durg treatment of PDAC organoids and xenografts” on page 16.

3. If combination of paclitaxel and gemcitabine is already known and reported, then authors should have focused on identifying if ETC-159 and combination of paclitaxel and gemcitabine can have increased efficacy. Authors can make an attempt of this combination. 

a. We thank the reviewer for this comment, seeing that triple combination was not specifically assessed through experiment in this study. While we did not look at the combination of all three drugs (ETC-159, paclitaxel, gemcitabine), we did assess both gemcitabine and paclitaxel as comparison to ETC-159 with each respective drug as a double combination. The double combination demonstrated more significant effect on the tumors than the combination of chemotherapy without ETC-159. Thus the triple combination would not have logically has an increased benefit, and this combination was not evaluated specifically in our pancreatic tumor models.

4. Colony formation can be represented as images

a. Thank you for identifying the images not included in the figures of the manuscript. In regards to using colony formation represented by images, initially we were going to take images and quantify by size. Through some preliminary data we found that to be too erratic, so we went to terminal cell viability assay to better quantify the colonies. Because of this, images were unable to be taken. We have included a figure of imaged of Panc269 and Panc272 to demonstrate why we went forward with CellTiter Glo 3D assays for quantification. This portion of the methods section as been clarified on page 8. 

5. Authors should carefully go through the figures and tables provided, the supplement figure 2 contains supplement table 2. 

a. Thank you for noticing errors in the figures of the manuscript and how they were referenced. We have thoroughly reviewed the manuscript and made changes to make it more clear to read, as well as going through the tables and figured provided to ensure they are accurate, clear, inclusive of all data referenced in the manuscript, and without duplicate.

6. Statistical analysis for all the graphs needs to be provided

a. Thank you for inquiring about the statistical analysis of the figures we have included in our manuscript. We have updated the graphs and statistics for Figure 1 and Figure 2. One-wayANOVA was utilized, and this was specified in figure legends, “Niche factor depletion studies for classification of Wnt (in)dependency of PDAC tumor organoids” section, and “Validation of Wnt dependency using Porcupine and Wnt inhibitors” section”. The remaining figured have statistical analysis done and included.

Reviewer #2: Hayley J Hawkins et al

Here are some constructive comments and observations you might consider for the provided based on the provided information about the research involving Examination of Wnt signaling as a therapeutic target for pancreatic ductal adenocarcinoma (PDAC) using a pancreatic tumor organoid library (PTOL), here are some questions you might consider:

I. The text provides a detailed and structured overview of the methods and procedures employed in the study.

II. The division into sections like "Human Specimens," "PDAC Patient-Derived Xenografts," and "Pancreatic Tumor Organoid Library (PTOL) Generation and Characterization" helps organize the information logically.

III. The process of generating PDAC organoid lines is clearly outlined, including details on tissue preparation, digestion, culture conditions, and media components.

IV. Assessment of Wnt Dependency:

a. the methodology for assessing the Wnt (in)dependency of PDAC tumor organoids is thoroughly explained, involving the manipulation of HPSC media and testing the effects on viability.

V. Wnt Inhibitor Studies:

a. The steps involving the use of Porcupine and Wnt inhibitors to confirm Wnt (in)dependency are well-documented, including the criteria used to define dependency.

VI. Minor Comments:

In certain sections, consider breaking down longer sentences for enhanced readability.

Thank you for recognizing ways that this manuscript can be simplified and be easier to read. We have addressed the recommendation to break down longer sentences to enhance readability. This has been done by breaking longer sentences into separate sentences to separate ideas. These changes can be found in the introduction on page 3, methods on pages 6-9, results on page 14, and discussion on pages 18-21.

For instance, consider using bullet points or subheadings for better organization in sections with multiple steps.

We again thank you for suggesting ways that the manuscript can be organized better. Sections with multiple steps were broken up into separate headings. The section titled “Pancreatic Tumor Organoid Library (PTOL) Generation and Classification” has been divided into separate sections for generation and classification on pages 5-6. The section titled “Assessment of the Wnt (in)dependency of PDAC tumor organoids” has been further divided into “HPSC media used for niche factor studies” and “Niche factor depletion studies for classification of Wnt (in)dependency of PDAC tumor organoids” on page 6-7.

The section on "Statistical Analysis of In vitro organoid drug treatment assays" is comprehensive but complex. Consider breaking down the information into subsections for clarity, separating details about individual drugs and combinations.

 We appreciate pointing out a specific section that would benefit from being structured differently. For the section on "Statistical Analysis of In vitro organoid drug treatment assays", we have separated this information into further sections according to individual drugs and analysis of anti-proliferative effects.

Clarify the notation "SMAD4(2, 3)." If these numbers represent specific mutations or isoforms, provide a brief explanation. Page 12 results section.

Thank you for addressing this need for clarification. The notation “SMAD4(2,3)” on page 12 is meant to just say SMAD4, followed by reference to citation. We have separated this to hopefully make it more clear, as well as separating the spacing for all references in the manuscript text. We have additionally added a sentence in the table legend of Table 1.

For better clarity, consider breaking the long sentence into two sentences: "Viability of Panc320 still required the presence of Noggin and A83-01. In contrast, Panc272 grew independently of all exogenous niche factors." Page 12

f. Clarify the significance or implications of the observed changes in gene expression in Panc320 and Panc269.

For better clarity, consider breaking the long sentence into two sentences: "Viability of Panc320 still required the presence of Noggin and A83-01. In contrast, Panc272 grew independently of all exogenous niche factors."

We see how this sentence could be broken up into two, and thank you for this suggestion. This has been addressed, and this specific sentence on page 14 has been broken up into two sentences. We have additionally revised the remainder for the manuscript to break up sentences where appropriate as specifically listed in section above.

Clarify the significance or implications of the observed changes in gene expression in Panc320 and Panc269.

Thank you for recognize these significances we were trying to communicate, and for advising us clarify such points. We have reviewed and clarified the significance of the observed changes as seen in Panc320 and Panc269 as discussed on pages 18-20, as well as clarifying in the opening sentence of the results section titled “Human Wnt pathway RT-PCR assay of post-treatment pancreatic tumors” that gene expression changes were observed to better understanding the mechanism behind Wnt (in)dependency, treatment susceptibilities, and treatment resistance.

---

## [Decision Letter · Decision Letter 1]

31 Jan 2024

Examination of Wnt signaling as a therapeutic target for pancreatic ductal adenocarcinoma (PDAC) using a pancreatic tumor organoid library (PTOL).

PONE-D-23-36221R1

Dear Dr. Hawkins,

We’re pleased to inform you that your manuscript has been judged scientifically suitable for publication and will be formally accepted for publication once it meets all outstanding technical requirements.

Kind regards,

Yash Gupta, Ph.D.

Academic Editor

PLOS ONE

Additional Editor Comments (optional):

Reviewers' comments:

Reviewer's Responses to Questions

**Comments to the Author**

1. If the authors have adequately addressed your comments raised in a previous round of review and you feel that this manuscript is now acceptable for publication, you may indicate that here to bypass the “Comments to the Author” section, enter your conflict of interest statement in the “Confidential to Editor” section, and submit your "Accept" recommendation.

Reviewer #1: All comments have been addressed

Reviewer #2: All comments have been addressed

2. Is the manuscript technically sound, and do the data support the conclusions?

Reviewer #1: Yes

Reviewer #2: Yes

3. Has the statistical analysis been performed appropriately and rigorously? 

Reviewer #1: Yes

Reviewer #2: Yes

4. Have the authors made all data underlying the findings in their manuscript fully available?

Reviewer #1: Yes

Reviewer #2: Yes

5. Is the manuscript presented in an intelligible fashion and written in standard English?

Reviewer #1: Yes

Reviewer #2: Yes

6. Review Comments to the Author

Reviewer #1: (No Response)

Reviewer #2: (No Response)

7. PLOS authors have the option to publish the peer review history of their article (what does this mean?). If published, this will include your full peer review and any attached files.

Reviewer #1: No

Reviewer #2: **Yes: **Manish Shukla

---

## [Editor Report · Acceptance letter]

8 Feb 2024

PONE-D-23-36221R1 

PLOS ONE

Dear Dr. Hawkins, 

I'm pleased to inform you that your manuscript has been deemed suitable for publication in PLOS ONE. Congratulations! Your manuscript is now being handed over to our production team.

Kind regards, 

on behalf of

Dr. Yash Gupta 

Academic Editor

PLOS ONE